# Embedded AMIS-Deep Learning with Dialog-Based Object Query System for Multi-Class Tuberculosis Drug Response Classification

**DOI:** 10.3390/diagnostics12122980

**Published:** 2022-11-28

**Authors:** Chutinun Prasitpuriprecha, Rapeepan Pitakaso, Sarayut Gonwirat, Prem Enkvetchakul, Thanawadee Preeprem, Sirima Suvarnakuta Jantama, Chutchai Kaewta, Nantawatana Weerayuth, Thanatkij Srichok, Surajet Khonjun, Natthapong Nanthasamroeng

**Affiliations:** 1Faculty of Pharmaceutical Sciences, Ubon Ratchathani University, Ubon Ratchathani 34190, Thailand; 2Department of Industrial Engineering, Ubon Ratchathani University, Ubon Ratchathani 34190, Thailand; 3Department of Computer Engineering and Automation, Kalasin University, Kalasin 46000, Thailand; 4Department of Information Technology, Buriram Rajabhat University, Buriram 31000, Thailand; 5Department of Computer Science, Ubon Ratchathani Rajabhat University, Ubon Ratchathani 34000, Thailand; 6Department of Mechanical Engineering, Ubon Ratchathani University, Ubon Ratchathani 34190, Thailand; 7Department of Engineering Technology, Ubon Ratchathani Rajabhat University, Ubon Ratchathani 34000, Thailand

**Keywords:** deep learning, AMIS, chest X-ray, drug-resistant, tuberculosis

## Abstract

A person infected with drug-resistant tuberculosis (DR-TB) is the one who does not respond to typical TB treatment. DR-TB necessitates a longer treatment period and a more difficult treatment protocol. In addition, it can spread and infect individuals in the same manner as regular TB, despite the fact that early detection of DR-TB could reduce the cost and length of TB treatment. This study provided a fast and effective classification scheme for the four subtypes of TB: Drug-sensitive tuberculosis (DS-TB), drug-resistant tuberculosis (DR-TB), multidrug-resistant tuberculosis (MDR-TB), and extensively drug-resistant tuberculosis (XDR-TB). The drug response classification system (DRCS) has been developed as a classification tool for DR-TB subtypes. As a classification method, ensemble deep learning (EDL) with two types of image preprocessing methods, four convolutional neural network (CNN) architectures, and three decision fusion methods have been created. Later, the model developed by EDL will be included in the dialog-based object query system (DBOQS), in order to enable the use of DRCS as the classification tool for DR-TB in assisting medical professionals with diagnosing DR-TB. EDL yields an improvement of 1.17–43.43% over the existing methods for classifying DR-TB, while compared with classic deep learning, it generates 31.25% more accuracy. DRCS was able to increase accuracy to 95.8% and user trust to 95.1%, and after the trial period, 99.70% of users were interested in continuing the utilization of the system as a supportive diagnostic tool.

## 1. Introduction

Tuberculosis (TB) is a deadly contagious disease that claims the lives of millions of people annually around the world. The World Health Organization (WHO) reports that, by 2020, it will have become the leading infectious agent-related cause of mortality [1]. The bacterium *Mycobacterium tuberculosis* that causes TB can be resistant to the antimicrobial medications that are used to treat it. Tuberculosis that does not respond to at least one first-line anti-TB drug is referred to as drug-resistant TB (DR-TB). Multidrug-resistant TB (MDR-TB) and extensively drug-resistant TB (XDR-TB) are two serious subtypes of DR-TB. Mignani et al. [2] proposed the suggested drug for each form of drug-resistant tuberculosis, which is a separate therapy regimen according to the patient’s drug-resistance status.

Both drug susceptibility testing (DST) results and the occurrence of serious adverse events are necessary for the development of a treatment strategy for tuberculosis (TB). Options for anti-TB treatment include first-line drugs, second-line drugs, and newer, more effective drugs (Figure 1).

The World Health Organization (WHO) recommends a 6-month-course of 2HRZ(E)/4HR for a patient with DS-TB as a standard treatment. If the patient’s clinical and radiographic status do not improve after 2 months of treatment with four drugs, DR-TB will be considered [4]. Multidrug-resistant tuberculosis (MDR-TB) is diagnosed when a patient is resistant to at least rifampin and isoniazid, the two most powerful first-line anti-TB drugs, and requires therapy with some second-line and new generation anti-TB treatments. Extensively drug-resistant tuberculosis (XDR-TB) is diagnosed when a patient does not react to the most effective second-line treatments, such as fluoroquinolones or injectable aminoglycosides, and requires multiple second-line and new generation anti-TB drugs [2].

The mismanagement of TB treatment and the dissemination of the disease from patient to patient are the main causes of the emergence and spread of drug-resistant TB. The treatment for tuberculosis typically consists in a medicine regimen to be taken daily for 6 months under close medical care. In crowded settings, such as prisons and hospitals, drug resistance is more likely to spread as a result of the improper or incorrect use of antimicrobial drugs, or the use of ineffective formulations of drugs (such as the use of single drugs, mediocre quality medicines, or bad storage conditions), and the premature interruption of treatment. Drug resistance can be diagnosed using specialized laboratory tests that examine the susceptibility of the bacteria to the medications, or that find patterns of resistance. These examinations may be molecular (such as Xpert MTB/RIF). Due to a lack of fully equipped laboratories, drug-resistance testing in a certain area can take a considerable amount of time; therefore, approaches that are both quick and cost-effective should be developed and used in this region. The gold standard for evaluating drug resistance is called drug susceptibility testing (DST). Beyond screening or triaging TB, a chest X-ray (CXR) is also used to recognize drug-resistant TB (DR-TB) due to the fact that CXRs are more convenient as a rapid, non-invasive, and cost-effective tool for TB diagnosis and therapy evaluation [5,6,7,8].

Machine learning (ML) and deep learning (DL) have become increasingly popular in recent years to categorize the response to anti-tuberculosis drugs due to their effectiveness and low-cost. Parkinson’s, breast cancer, and chronic kidney disease are only a few examples of diseases where ML and DL have been used to accurately forecast or estimate outcomes. Automated DL-based methods that rely on CXRs for tuberculosis (TB) diagnosis have been shown to be highly effective [9,10]. Karki et al. [11] used radiological parameters and clinical patient data in conjunction with chest X-ray images to categorize tuberculosis as drug-resistant (DR) or drug-sensitive (DS).

Machine learning was ultimately used by Tulo et al. [12] to categorize four distinct types of drug-resistant TB. This includes DS-TB, DR-TB, MDR, and XDR, which some refer to forms of tuberculosis that have developed resistance to at least some of the available treatments. A binary classification approach was used to categorize these four types of medication responses. The computational results show that there is room for improvement in the accuracy of the binary classification between DR-MDR, MDR-XDR, and DR-XDR, which lie between 82% and 93%. When diagnosing the TB medication response in the real world, there should be a single system that can classify all four types of drug reactions using a single CXR picture input, since it would be more convenient for clinicians to utilize.

Multi-class classification is a classification challenge in deep learning and machine learning that involves more than two classes or results. Images from the TB dataset require multi-class classification since there are at least four distinct treatment responses that must be recognized by the model. It is more challenging to predict the outcome for multi-class classification than for binary class classification since multiple outcomes must be gathered from the same dataset. Bayesian classifiers that employ Bayes’ theorem [13], the decision tree model [14,15], and the ensemble deep learning models [11] are all examples of methods that could be improved upon to create a more effective classification process.

After carefully reviewing the aforementioned literature, we have developed our study to target the following areas that have been overlooked to date:The classification should be performed as four classes, which are DS, DR, MDR, and XDR, in order to suggest the correct recommended treatment program to the treaters.The designed model has not yet been used as the suggestion system, such as mobile application or chatbot, in order to use the design model in real medical treatments.The lowest accuracy is still 82%, which is not significantly high and can result in a lack of trust from doctors.

Inspired by these unmet needs, we propose the design of a comprehensive classification system for drug resistance that distinguishes between DS, DR, MDR, and XDR. To achieve this, the system will be developed using the following techniques: (1) Create an effective multi-class classification model, and (2) create an application utilizing the model created in (1). The remaining sections are organized as follows. Section 2 will review the related work, Section 3 will describe the materials and methods used in this study, and Section 4 and Section 5 will present the computational results and discussion. The investigation’s conclusions will be reported in Section 6.

## 2. Related Work

The theory and related literature will be divided into two major sections: (1) The effective classification methods, and (2) the dialog-based object query system (DBOQS) development system.

### 2.1. The Effective Classification Methods

#### 2.1.1. Chest X-ray and Deep Learning

Karki et al. [11] and the World Health Organization [16] proposed a method for classifying DS-TB and DR-TB using a system of deep learning. The CNN architecture ResNet18 was utilized in this research. Radionics [17] has been utilized to identify patterns that radiologists may have overlooked when examining anomalies in medical images. Multi-task learning [18] was employed to standardize the core model and to guide the network’s focus on the seemingly relevant portions of the image. To prepare the CXR image for processing, the Original CXR is provided to U-Net, which generates a binary lung mask. The Original CXR is then cropped, and the lungs are segmented inside the bounding box that has been diminished. Tulo et al. [12] used machine learning (ML) to examine the lungs and mediastinum, in order to classify the TB patients’ drug response. Despite the use of binary categorization [19] in this study, other kinds of drug-resistant tuberculosis, including DS, MDR, and XDR, are predicted. Several machine learning algorithms, including multi-layer perceptron [20,21], K-nearest neighbor [22,23], and support vector machine [23,24] were used to identify the solution.

Successful deployments of deep learning architectures have been carried out in many fields, from image/video classification to healthcare [25]. Due in large part to the automatic parsing of radiology reports to generate labels, the CXR research community has profited in recent years from the publication of multiple large, labeled databases, satisfying the insatiable need for data that are inherent to deep learning. In 2017, the NIH clinical center began this pattern by releasing 112,000 photos [26]. Additional articles later demonstrated that deep learning is superior to other forms of AI in CXR analysis [27,28,29].

#### 2.1.2. Ensemble Deep Learning Model

Ensemble deep learning is a deep learning architecture that uses multiple architectures to address a problem within a single model. Combining the predictions of numerous models has proven to be an excellent way to improve the performances of models. Ensemble learning or ensemble model refers to the process of mixing many predictions from various models, in order to arrive at a final solution. The solution quality of deep learning is contingent on many mechanisms, such as data preparation techniques, deep learning architectures, and fusion techniques.

##### Image Preprocessing Method

The chest X-ray images used in this research came from seven different countries: Belarus, Georgia, Romania, Azerbaijan, India, Moldova, Ukraine, Kazakhstan, and South Africa. Diverse X-ray machines produce vastly different results in terms of image quality. The initial dimensions of the amassed images varied in the number of pixels. The image analysis workflow highly depends on the pre-processing phase. Through this, it is possible to improve the original image while simultaneously decreasing the amount of noise or distracting elements. Easy image preprocessing methods can dramatically increase the qualities of the classification results of ML and DL. Karki et al. [11] employed image segmentation to restrict the input pictures for the binary DR/DS classifier to only include regions that are significant for the categorization of pulmonary tuberculosis, i.e., the lungs. In addition, potential confounding elements are eliminated and the lungs are resized in order that they occupy the same percentage of the image. Lung size and patient placement, for example, are commonly connected with clinical settings and could serve as confounding variables. After the segmentation is completed, the results reveal that their proposed model benefits greatly from including these techniques.

Caseneuve et al. [30] introduced two methods that are referred to as the threshold approach and edge detection to identify CXR images with two clearly discernible lung lobes. These two methods were founded on the concepts of Otsu’s threshold [31] and the Sobel/Scharr operator [32,33]. Sobel examines the presence of the upper/lower image boundaries or left/right image boundaries in the horizontal and vertical gradient calculations. Specifically, the Sobel operator will indicate if visual changes are rapid or gradual. The Scharr operator represents a modification where the mean squared angular error is measured as an extra optimization and an introduction of greater sensitivity. This enables the kernels to be of a derivative type, which approximates Gaussian filters more closely.

Before feeding the images into the CNN model, Ahamed et al. [34] and Wang et al. [35] applied the cropping and sharpening filters of image processing techniques to the obtained datasets to enhance the quality of the images, to extract the primary portion of the lung image, i.e., to remove undesirable, irrelevant portions of the photos. The images were trimmed with the correct height-to-width ratio in mind. All of the acquired photos were then filtered with a sharpening filter for enhancement. Conceptually, this filter is derived from Laplacian filters, and is an example of a second-order or second-derivative system of enhancement that accentuates regions of fast intensity variation [36,37].

One of the most popular preprocessing techniques that are used to improve the image quality is contrast enhancement. Many contrast enhancement techniques have been introduced to improve the contrast of an image, such as the histogram equalization (HE)-based methods. The classical HE can efficiently utilize displayed intensities, but it tends to over-enhance the contrast if there are high peaks in the histogram, which often results in a harsh and noisy appearance of the output image. Numerous methods have been proposed for limiting the level of enhancement, most of which are obtained through modifications on HE [35,38].

In Arici et al. [39], the authors present a histogram modification framework (HMF) that treats contrast improvement as a minimization of a cost function. To deal with noise and black/white stretching, penalty terms are applied during optimization. Manually adjusting the parameters of HMF allows for varying the degrees of contrast improvement. An automatic image enhancement method based on gamma correction has recently been developed [40]: Contrast enhancement using adaptive gamma correction with weighting distribution (AGCWD).

If there are high peaks in the input histogram, the AGCWD may be overly simplistic and lead to image blurring in the bright areas. When applied to the entire input image, these global methods are effective, but they overlook the local details. Approaches based on local histogram equalization (LHE) are presented as a means of resolving these shortcomings. The contrast limited adaptive histogram equalization (CLAHE) method proposed by Pizer et al. [41], is a classic LHE-based image enhancement method that first divides the image into a large number of continuous and non-overlapping sub-blocks, then enhances each sub-block independently, and finally employs an interpolation operation to reduce the block artifacts.

While extreme contrast is possible with the LHE, the final image may be too bright. Furthermore, LHE-based approaches typically necessitate a greater computational effort than GHE-based approaches. Another issue with LHE-based approaches is the dramatic increase in background noise [41,42,43,44]. The second class of contrast enhancement methods conducts image decomposition before performing enhancement, in order to reduce artifacts and increase subjective quality. Lee et al. [45] suggested that a gradient domain tone mapping technique can be used. Artifacts may be introduced in non-integrable gradient fields when obtaining the improved image by solving a Poisson equation on the altered gradient field. Edge-aware image processing using the Laplacian pyramid (LPM) was presented by Paris et al. [46].

##### CNN Architectures

In addition to the data de-noising and augmentation approaches that may be utilized to improve the classification quality of DL, the convolution neural network (CNN) architecture is one of the most essential characteristics that can considerably increase the classification solution quality. In recent years, numerous CNN architectures have been proposed to improve picture classification performance. CCNet [47], VGG16, VGG19 [48], ResNet50 [49], ResNet101 [50], DenseNet121 [51], MobileNetV2 [28], EfficientNetB1 [52], NASNetMobile [53], and EfficientNetB6 [54] are examples of effective CNN state-of-the-art designs.

Although these types of structures have not yet been applied to the classification of drug-resistant organisms, they have been utilized in a variety of applications. ResNet101 successfully solved the detection of gravelly soil uniformity, and it was superior to all other architectures [55]. MobileNetV2 is a small, low-latency, low-power model parameterized to satisfy the resource restrictions of a range of use cases. For instance, Zhang et al. [56] successfully employed MobileNetV2 to determine the fish school feeding behavior. NASNetMobile was proposed by Maharjan [57] to detect COVID-19 infection in posteroanterior chest X-rays, whereas Chaganti [58] and Hoorali [59] used EfficientNetB1 and EfficientNetB6 to classify malware and segment medical images, respectively. Although the abovementioned designs have not yet been used to solve DR-TB and its associated types of DR, it has been demonstrated in numerous publications that they are effective solutions. In this article, a modified version of MobileNetV2, NASNetMobile, EfficientNetB7, and DenseNet121 will be utilized to determine the TB’s drug response.

##### Decision Fusion Strategies

The effective fusion methodologies utilized to merge the results of several CNN architectures will be employed to improve the solution quality of the TB classification. Decision fusion techniques are the processes that occur when ensemble learning trains many base learners and aggregates their outputs using specified rules. Effective ensemble performance is determined by the rule used to aggregate the outputs. The majority of ensemble models concentrate on the ensemble architecture, and then use naive averaging to forecast the ensemble output. However, the naive averaging of the models, which is followed by the majority of ensemble models, is not data-adaptive, and results in a sub-optimal performance [24] since it is susceptible to the performances of biased learners.

As there are billions of hyper-parameters in the architecture of deep learning, the issue of overfitting may result in the failures of some base learners. Consequently, techniques, such as the Bayes optimum classifier and super learner, have been adopted to address these problems [24]. In the literature, the many methods for merging the outputs of ensemble models are as follows: (1) Unweighted model averaging, (2) majority voting, (3) Bayes optimum classifier, (4) stacked generalization, (5) super learner, (6) consensus, and (7) query-by-committee. The unweighted model average weight of all CNNs employed has the same weight, while majority voting uses the decisions of all CNNs to count and select the outcome that the majority of CNNs decide to forecast. Meanwhile, the remaining technique attempts to identify the optimal weight used to make the model’s decision.

Recently, Gonwirat and Surinta [60] published the differential evolution (DE) approach for automatically optimizing the weighted parameter discovery. The DE algorithm is applied to the weighted parameters, and the optimal weight is then assigned to the ensemble method and stacked ensemble technique. The outcome reveals that the optimal weight finding technique utilizing DE provides a superior answer to the unweighted model averaging and majority voting. Pitakaso et al. [61] proposed the novel heuristics artificial multiple intelligence system (AMIS) to deal with the network flow problem, and the results suggested that AMIS outperformed DE in locating the superior solution. In this study, we will employ AMIS-ensemble deep learning (AMIS-EDL) to discover the appropriate weight to use as the decision fusion approach rather than the conventional approach.

### 2.2. DBOQS in Healthcare

Powered by artificial intelligence (AI), DBOQSs are gaining popularity in a variety of businesses, and have substantial application potential in real-world scenarios. However, medical DBOQSs have received little attention, with the majority of published evidence focusing on technology challenges and with limited application research. Moreover, healthcare service providers are interested in adopting new technologies to improve their services [62], and are beginning to adopt medical DBOQSs for answering/asking questions, creating health records and histories of use, providing information about diseases, discussing the results of clinical tests, and even taking appropriate actions based on users’ responses [63,64]. Typically, the DBOQS has been utilized to communicate with patients, in order to maintain the doctor–patient relationship, such as with cancer [65], mental health [66,67], and COVID-19 patients [68]. It has been demonstrated that the use of DBOQSs decreases the workload of medical personnel and increases patient satisfaction in healthcare [69].

The DBOQS controller was designed to manage mobile communication between the three CNNs and the users. To detect the health of various crops, including pomegranate trees and firecracker plants, Jain et al. [70] utilized a CNN with a customized architecture hosted on a cloud service and a mobile application for Android smartphones. Picon et al. [71] and Esgario et al. [72] found three wheat illnesses and four coffee leaf diseases and pests, respectively, using a mobile application and a CNN with a modified RetNet-50 architecture hosted on a cloud service. Therefore, the mobile DBOQS is applicable to a variety of real-world issues. Temniranrat et al. [73] were able to detect five common rice diseases using a CNN with YOLOv3 architecture hosted on a cloud service and LINE, an instant messaging application that is maintained by its development team [74]. A DBOQS was placed on an instant messaging application (LINE) to access CNNs hosted on a cloud service; therefore, the system offered automatic responses and was always accessible.

From the vast amount of literature, we can conclude that it is possible to use DBOQSs in the field of healthcare image processing, particularly in TB medication response classification, where they only receive an image as input and then report the result using a deep learning model embedded in DBOQS. Using an online DBOQS to inquire about the CXR owner’s chance of developing DR can alleviate the doctors’ workload. Furthermore, this study will provide the recommended treatment plan for the selected patients.

## 3. Materials and Methods

This study seeks to improve the precision of the deep learning technique used to classify tuberculosis patients’ medication responses. The drug reaction will be divided into four classes, namely DS, DR, MDR, and XDR. After obtaining an effective algorithm to classify them, the DBOQS will be utilized to converse with the doctor, in order to recommend the situation of the patients and the treatment plan for those patients. The therapy schedule recommended by the WHO is indicated in Figure 1. Consequently, the study approach, depicted in Figure 2, will be implemented.

The method used to develop the DRCS consists in three steps: (1) Collecting the dataset of the CXRs, as well as the efficacies of previous methods from various literatures; (2) developing the AMIS-EDL algorithm to build the model to classify the drug response; and (3) developing the DRCS using DBOQS. The following is a detailed explanation of each step.

### 3.1. The Revealed Dataset and Compared Methods

In this experiment, the same dataset was utilized by Karki et al. [11]. The data collection contains 5019 CXR images associated with tuberculosis, of which 3412 are from DR-TB and 1607 are from DS-TB. It is accessible for download at https://tbportals.niaid.nih.gov (accessed on 2 August 2022) [75]. The vast bulk of TB portal data were obtained to identify DR-TB patients from a diverse sample of TB cases. To evaluate the performance, the dataset was partitioned for training and testing at 80% (*n* = 4015) and 20% (*n* = 1015), respectively. Table 1 displays the number of DS-TB, DR-TB, MDR-TB, and XDR-TB data used for training and testing.

Table 1 indicates that the total numbers of DS-TB, DR-TB, MDR-TB, and XDR-TB have 1607, 468, 2087, and 857 datasets, respectively; or 32.02%, 9.32%, 41.58%, and 17.08%, respectively. These data will be used to train and test for the accuracy of the proposed methods, compared with the methodology proposed by Ureta and Shrestha [76], Tulo et al. [77], Jaeger et al. [24], Kovalev et al. [78], Tulo et al. [12], and Karki et al. [11]. The details of the compared method are presented in Table 2.

The key performance indicators (KPIs) to evaluate the suggested methods in comparison to other methods include area under the curve (AUC), F-measure, and accuracy. For binary classification issues, an evaluation metric is the receiver operator characteristic (ROC) curve. The area under the curve (AUC), which serves as a summary of the ROC curve, is a measurement of a classifier’s capacity to distinguish between classes. The harmonic mean of recall and precision is used to calculate the F-measure, giving each the same weight. It enables the evaluation of a model, taking into consideration both precision and recall using a single score, which is useful for explaining the performance of the model and when comparing models. The model’s performance across all classes is often described using its accuracy metric. When every class is equally important, it is useful. It is determined by dividing the total number of guesses by the number of predictions that were correct.

### 3.2. The Development of Effective Methods

In this study, three methodologies will be utilized to increase the efficacy of the classification of the TB drug response. The data preparation techniques, modern CNN architectures, and decision fusion techniques are these methodologies. Each can be described in greater detail, as follows.

#### 3.2.1. The Data Preprocessing Method

Data preprocessing involves transforming or encoding data in order that they may be easily parsed by a machine. For a model to make accurate and exact predictions, its algorithm must be able to easily interpret the data’s characteristics. Due to their varied origins, the bulk of real-world datasets for machine learning are very prone to be missing, inconsistent, and noisy. The application of data preparation algorithms to these noisy data would not yield excellent results, since the algorithms would be unable to find patterns adequately. Two types of image preprocessing algorithms will be adapted for usage with CXRs, in order to improve the classification accuracy of the proposed model. These two methods are data augmentation and data normalization.

##### Data Augmentation

By creating synthetic datasets, data augmentation aims to increase the quantity and variety of the training data. The augmented data can be considered to have been derived from a distribution that closely resembles the actual distribution. Then, the expanded dataset can represent features that are more thorough. Image data augmentation techniques can be applied to a variety of data types, including object identification [79], semantic segmentation [80], and picture classification [81].

Image manipulation, image erasure, and image mixing are the fundamental image augmentation techniques. In this research, only image manipulation will be used. Image transformations, such as rotation, mirroring, and cropping, are the focus of fundamental image transformations. The majority of these methods modify images directly, and are simple to implement. Nevertheless, there are downsides. First, it only makes sense to apply fundamental image modifications if the existing data follow a distribution that is near the actual data. Second, some basic image manipulation techniques, such as translation and rotation, suffer from the padding effect. Specifically, following the operation, some sections of the images will be shifted outside the boundary and lost. Consequently, various interpolation methods will be used to fill in the missing data. Typically, the region outside of the image’s border is presumed to be a constant of 0, and will be dark following alteration. Figure 3 depicts an example of image enhancement employing several types of enhancement, as described earlier.

##### Image Normalizing Algorithm

Four steps of normalizing the CXR input data have been used. These four steps are shown in Figure 4.

In Figure 4, the Original CXRs will be processed with adaptive masking (AM), Gaussian blur (GB), CLAHE, and MVSR. AM is the method depicted in Heidari et al. [82]. It can execute the procedure by removing the sample’s diaphragm. AM begins by determining the maximum (max) and minimum (min) pixel intensities, followed by the use of threshold techniques for binary thresholding and morphologic closure. This generates the adaptive mask that eliminates the aperture from the source image after a bitwise operation. Gaussian blur (GB) is a filter that operates by calculating a pixel’s value. The filter is based on the normal distribution, which has the form of a bell curve. The concept is that pixels closest to the center pixel carry a greater weight than those further away.

The CLAHE algorithm comprises three main components: Tile generation, histogram equalization, and bilinear interpolation. First, the input image is segmented into pieces. Each unit is referred to as a tile. The input image depicted in the illustration is separated into four tiles. Then, each tile is subjected to histogram equalization using a clip limit that has been specified. Histogram equalization involves five steps: Histogram computation, excess calculation, excess distribution, excess redistribution, and scaling and mapping with a cumulative distribution function (CDF). For each tile, a collection of bins is used to compute the histogram. Histogram bin values exceeding the clip limit are gathered and dispersed over other bins. Then, the CDF for the histogram values is calculated. Using the supplied image pixel values, each tile’s CDF values are scaled and mapped. The generated tiles are stitched together using bilinear interpolation to produce an image with enhanced contrast.

Mean-variance-Softmax-rescale normalization (MVSR normalization) is based on four mathematical operations: The mean of the data, variance, Softmax, and rescaling. According to the probability theory of R. Duncan Luce, sometimes known as Luce’s choice axiom, the probability of one sample being within the same dataset depends on another sample. Typically, the Softmax function is employed in the final activation function of multi-class. In general, the standard deviation (the square root of the variance) is used to create a link between data points and to quantify the spread or distribution of a data collection relative to its mean [83], using artificial neural network (ANN) models to build an output class probability distribution [84]. After calculating the normalized intensity of the input, the dataset may include both negative and positive fractional values. The Softmax function was utilized to maintain the impact of negative data and nonlinearity.

To normalize the raw input data into a higher quality of image for use as the input data, in order to increase the performance of the TB classification, image normalization is a crucial component of the image analysis schema. It can improve the original image by minimizing noise or unnecessary elements. In our research, we aligned the training and testing photos by enhancing low-contrast, high-noise input images using four distinct normalization techniques to increase contrast and sharpness. Figure 5 is an example of when the proposed preprocessing methods are applied.

#### 3.2.2. CNN Architectures

In this research, we will use four compact CNN architectures in an effort to limit the amount of time spent on model construction. As a result, we will use four compact CNN architectures that are nonetheless rather powerful.

##### MobileNetV2

Sandler et al. [28] introduced MobileNetV2, which uses depthwise separable convolutional (DwConv) layers and the inverted residuals of the bottleneck block to reduce the weighted parameters of a lightweight network. Depthwise separable convolution is used as a foundational building block in MobileNetV2. However, it adds linear bottlenecks between the layers and shortcut links between the bottlenecks to the design. Figure 6 depicts the structure of the MobileNetV2’s system.

The bottlenecks encode the model’s intermediate inputs and outputs, while the inner layer embodies the model’s capacity to change from lower-level ideas, such as pixels to higher-level descriptors, such as image categorizations. Finally, as in the case with conventional residual connections, shortcuts allow for faster training and greater precision. MobileNetV2 models are faster over the whole latency spectrum with the same precision. Specifically, the new models use two-fold operations, require 30% fewer parameters, and are around 30% to 40% faster on a Google Pixel phone than MobileNetV1 models, all while attaining more accuracy [28].

##### EfficientNetB7

EfficientNet was created by Tan and Le [52] to search the hyper-parameters of CNN architectures, such as width scaling, depth scaling, resolution scaling, and compound scaling. In addition, squeeze-and-excitation (SE) optimization was added to the bottleneck block of EfficientNet, in order to generate an informative channel feature with GAP summation. Then, correlation features are identified by lowering the dimensions to small sizes and changing them back to their original size. EfficientNetB1 was developed based on MobileNetV2; however, its resolutions, channels, and repetition rates vary. EfficientNetB1 is comparable to MobileNetV2, which replaced Conv1 with 512 feature maps and eliminated the fourth bottleneck block. Figure 7 depicts the architecture of the EfficientNetB1 overall architecture, which may be separated into seven blocks. Each block of MBConv’s associated filter size is displayed.

##### DenseNet121

In a conventional feed-forward convolutional neural network (CNN), each convolutional layer, with the exception of the first, obtains the output of the previous convolutional layer and generates an output feature map that is then transferred to the next convolutional layer. Therefore, for “L” layers, there are “L” direct connections, one between each layer and the following one. However, when the number of layers in the CNN increases, i.e., as they become deeper, the “vanishing gradient” issue develops. This implies that as the channel for information from the input to output layers becomes longer, certain information may “disappear” or become lost; therefore, reducing the network’s capacity to train successfully. DenseNets solve this issue by changing the traditional CNN design and streamlining the interlayer connectivity structure; therefore, the moniker “densely connected convolutional network”. L(L + 1)/2 direct connections exist between “L” levels. In each layer, the feature maps of all preceding layers are not added together, but rather, they are concatenated and used as inputs. As a result, DenseNets require fewer parameters than a comparable standard CNN, which enables feature reuse as redundant feature mappings are removed.

When the sizes of feature maps vary, it is not possible to use the concatenation method. Nevertheless, a crucial component of CNNs is the down sampling of layers, which minimizes the size of feature maps via dimensionality reduction, in order to increase computation speeds. To achieve this, DenseNets are subdivided into DenseBlocks, where the dimensions of the feature maps remain constant inside a block, but the number of filters between blocks varies. Transition layers are the layers between the blocks that lower the number of channels by half. For each layer, the above equation is solved.

An example of a deep DenseNet with three dense blocks is displayed in Figure 8; within the dense block, the feature maps are all the same size, in order that the features can be concatenated, while the transition layers between neighboring blocks conduct down sampling via convolution and pooling procedures. DenseNet121 comprises 7 × 7 convolutions and 3 × 3 max pooling. It has $⎣\begin{array}{c} 1\times 1\\ 3\times 3 \end{array}⎦\times 24$ dense blocks (3) and $⎣\begin{array}{c} 1\times 1\\ 3\times 3 \end{array}⎦\times 16$ dense blocks, (4) while for other dense blocks, it has the same as DenseNet169, DenseNet201, and DenseNet264.

##### NASNetMobile

The Google brain team created the neural architecture search network (NASNet), which has two key functions: Normal cell and reduction cell. In order to achieve a higher map, NASNet initially applies its operations to the small dataset before transferring its block to the large dataset. For optimal regularization, the speed of NASNet is increased via a customized drop path known as the scheduled drop path. In the original NASNet architecture, where the number of cells is not predetermined, normal and reduction cells are used [85], with the normal cells determining the feature map size and the reduction in the cells returning the feature map with its height and width decreased by a factor of two. A recurrent neural network (RNN)-based control architecture in NASNet predicts the entire network topology using two initial hidden states. The controller architecture uses an RNN-based LSTM model with Softmax prediction for convolutional cell prediction and recursively produced network motifs. NASNetMobile accepts images with a resolution of 224 by 224 pixels, whereas NASNetLarge accepts images with a resolution of 331 by 331 pixels. NASNetMobile leverages the pre-trained ImageNet network weights for transfer learning, in order to recognize the objects. Figure 9 shows the architecture of NASNetMobile.

All the architectures that are previously explained will be used in both heterogeneous and homogeneous manners in the ensemble deep learning, as shown in Figure 10.

#### 3.2.3. Decision Fusion Strategy

The decision fusion strategy (DFS) combines the classification decisions of many classifiers into a single conclusion about the event. When it is challenging to combine all of the CNN’s results, several classifiers are frequently employed with multi-modal CNN. Several viable DFSs for application in deep learning ensembles, including the unweighted model average, majority voting, and various weight optimization techniques, were discussed in [86]. The designed methodologies for this investigation will employ three approaches. These methods are the unweighted model average and majority voting, as described in Johnson et al. [87], together with our proposed DFS, which is the AMIS-DFS developed by Pitakaso et al. [61].

##### Unweighted Model Average

The most prevalent method for fusing decisions in the literature is the unweighted average of the outputs of the base learners in an ensemble. The final prediction of the ensemble model is obtained by averaging the results of the base learners. Since deep learning architectures have a high variance and a low bias, a simple averaging of the ensemble models improves the generalization performance by reducing the variance among the models. The averaging of the base learners is conducted directly on the outputs of the base learners, or on the predicted probabilities of the classes using the Softmax function [88] illustrated in Equation (1).
(1)$$P_{i}^{j}=softmax^{i}\left(O_{I}\right)=\frac{o_{I}^{J}}{\sum_{k=1}^{K}exp\left(o|K^{j}\right)}$$
where $P_{i}^{j}$ is the probability of the *i*-th unit on the *j*-th base learner, $o_{I}^{J}$ is the output of the ith unit of the *j*-th base learner, and *K* is the number of classes.

##### Majority Voting

Similar to unweighted averaging, majority voting aggregates the base learners’ outputs. In contrast, majority voting counts the votes of the base learners and predicts the final labels as the label that received the majority of votes. In contrast to unweighted averaging, majority voting is less skewed towards the outcome of a certain learner, since the influence is offset by a majority vote count. However, if the majority of similar or dependent base learners favors a certain event, that event will predominate in the ensemble model. In majority voting, Kuncheva et al. [89] found that pairwise dependence among the base learners plays a significant influence, whereas for the categorization of images, the prediction of shallow networks is more diversified than those of deeper networks [90].

##### Optimization of Weight Using AMIS-DFS

The principle underlying AMIS is to use a system with many artificial intelligences to aid in discovering optimal solutions. “Intelligence box or IB” refers to a collection of computing methods or algorithms with unique properties. AMIS is a heuristic based on population. A population member is known as a work package (WP). Each WP will independently enhance its solution using the chosen IB. AMIS will generate a baseline set of work packages (WP), WP will select IB to improve its own solution in each iteration, update its heuristics’ information, and repeat this cycle until the termination condition is met. The AMIS can be elaborated upon as follows.

##### Initial Work Package Generation

At random, initial work packages (WP) will be generated. In this phase, the NP number of WPs is displayed. The number of CNN defines dimension D of the WP. The first set of WP is illustrated in Table 3. Let *X_ijt_* stand for the first set of *WP_i_* (*i* = 1 in this case), element *j* iteration *t*.

At iteration *t*, the weight that will be used to classify the drug response for a CXR can be determined using Equation (2). Figure 11 depicts the framework for calculating the total weight using the value of element *j* of *WP_i_* at iteration *t*.

Let $Y_{AVG}$ be the average value of the prior classification value for a specific *i* and *t*. $Y_{AVG}$ can be determined with Equation (2).
(2)$$Y_{AVG}=\sum_{j=1}^{J}W_{j}Y_{j}$$

When $W_{j}$ equals $X_{ijt}$ as determined via the AMIS procedure, *i* is the number of WP, *j* is the number of CNN or the number of WP elements, and *t* is the iteration counter.

##### Performing the WP Execution Process

In this phase, the WP will select the improvement box (IB) to enhance the quality of the current solution, executing the WP improvement process repeatedly without relying on past results. During each iteration, the WP is accountable for selecting the IB that will increase the quality of its solution. In this study, the following intelligence boxes are utilized. Equations (3) through (10) are used to perform the various IBs. When it is time to execute all WPs in iteration t, each WP will choose its preferred intelligence box (IB) using Equation (3). Equation (3) is based on the idea that the quality of earlier solutions to WPs influences the desirability of choosing the IB.
(3)$$Z_{ijt}=\rho X_{rjt}+F1\left(B_{j}^{gbest}-X_{rjt}\right)+F2\left(X_{mjt}-X_{rjt}\right)$$
(4)$$Z_{ijt}=X_{rjt}+F1\left(B_{j}^{gbest}-X_{rjt}\right)+F2\left(B_{hj}^{pbest}-X_{rjt}\right)$$
(5)$$Z_{ijt}=X_{rjt}+F1\left(X_{mjt}-X_{njt}\right)$$
(6)$$Z_{ijt}=X_{rjt}+R_{ij}\left(X_{rjt}-X_{njt}\right)$$
(7)$$Y_{ijt}=R_{ij}$$
(8)$$Z_{ijt}=\{\begin{array}{c} X_{ijt}ifR_{ij}\leq CR\\ R_{ijt}otherwise \end{array}$$
(9)$$Z_{ijt}=\{\begin{array}{c} X_{ijt}ifR_{ij}\leq CR\\ X_{njt}otherwise \end{array}$$
(10)$$Y_{Zjt}=\{\begin{array}{c} X_{ijt}ifR_{ij}\leq CR\\ R_{ij}X_{ijt}otherwise \end{array}$$

$X_{ijt+1}$ is the value of WP *i* element j in iteration *t* + 1, whereas r, m, and n are members of the set of WP (1 to *I*) that are not equal to *i*. $B_{j}^{gbest}$ is the global best WP, which has the best solution compared to all other created WPs. *R_ij_* is a random number of WP *i* element *j*. *F1* and *F2* are the scaling factors, which are defined as 3 (as recommended by the result of [61]), and CR is the crossover rate, which [61] suggests is equal to 0.8. *R_ijt_* is the randomly produced WP of WP *i* element *j* in iteration *t*.
(11)$$P_{bt}=\frac{FN_{bt-1}+\left(1-F\right)A_{bt-1}+KI_{bt-1}}{\sum_{b=1}^{B}FN_{bt-1}+\left(1-F\right)A_{bt-1}+KI_{bt-1}}$$

For each iteration *t*, the probability to select IB b is denoted by *P_bt_*. Assume that *N_bt_*_−1_ is the number of WP picking *IB b* in the previous iterations; *A_bt_*_−1_ is the average objective value of all WP picking *IB b* in the previous iterations, *I_bt_*_−1_ is a reward value that increases by 1 if *IB b* finds the optimal solution in the last iteration and an added value of 0; otherwise, B is the total number of *IB*, *F* = 2 is the scaling factor, and *K* = 1 is the improvement factor. *X_ijt_*_+1_ was updated using Equation (12).
(12)$$X_{ijt+1}=\{\begin{array}{c} Z_{ijt}iff\left(Z_{ijt}\right)\leq f\left(X_{ijt}\right)\wedge updatef\left(X_{ijt}\right)=f\left(Z_{ijt}\right)\\ X_{ijt+1}otherwise \end{array}$$

##### Updating the Heuristics Information

Some heuristics data need to be updated before they can be used in subsequent iterations. The new regulation is shown in Table 4.

The work package must be carried out in full until the prerequisites for its completion have been met.

Repeat the step from WP generation to heuristic information update until the prerequisites for halting have been met using this step. In this case, the stopping criterion is the elapsed processing time or the maximum number of iterations. The AMIS pseudocode is shown in Algorithm 1.
**Algorithm 1:** Artificial Multiple Intelligence System (AMIS)input: Population size (NP), problem size (D), mutation rate (F), recombination rate (R), number of intelligence box (NIB)output: Best_Vector_Solution**begin**   *Population = Initialize set of WPs*   *IBPop = Initialize InformationIB (NIB)*   *encode Population to WP*  ***while***
*the stopping criterion is not met **do***  ***for** i = 1: NP*    *//selected Intelligence box by RouletteWheelSelection*           *Selected_IB = RouletteWheelSelection (IBPop)*    ***if** (selected_IB = 1) then*           *new_u = using Equation (3)*    ***else if** (selected_IB = 2)*           *new_u = using Equation (4)*    ***else if** (selected_IB = 3)*           *new_u = using Equation (5)*    ***else if** (selected_IB = 4)*           *new_u = using Equation (6)*    ***else if** (selected_IB = 5)*           *new_u = using Equation (7)*    ***else if** (selected_IB = 6)*           *new_u = using Equation (8)*    ***else if** (selected_IB = 7)*           *new_u = using Equation (9)*    ***else if** (selected_IB = 8)*           *new_u = using Equation (10)*    ***if** (CostFunction(new_u)) ≤ (CostFunction (V_i_)) then*           *V_i_ = new_u*    *//Loop for updating the intelligence box’s heuristics data*    ***for** j = 1: NIB*           *interpreting WP to discover the real problem*    ***end** For Loop-end update heuristics information*   ***end** for Loop*  ***end*** ***return** Best_Vector_Solution* ***end***

Figure 12 demonstrates that when data are obtained from their source, they will be preprocessed using two types of preprocessing methods: (1) Image augmentation methods and (2) data normalizing methods. Thereafter, four types of CNN architectures will be used as the EDL tool. These are EfficientNetB7, DenseNet121, MobileNetV2, and NASNetMobile. Finally, the model will employ three decision fusion strategies: (1) Unweighted model averaging (UMA), (2) majority voting (MV), and AMIS-EDL. Therefore, 60 proposed models will be generated, as shown in Table 5.

### 3.3. DBOQS Design

The DBOQS was developed to provide the medical team with an answer to the question of whether or not the patient has the probability of receiving the DR. Figure 13 depicts the framework of the DBOQS that we designed for this project.

From Figure 13, when a doctor (user) wants to use the DBOQS to inquire about the drug response of the TB patients, the doctor must enable the LINE Application on their mobile phone to scan the QR code to add DBOQS as a friend for the first time. Then, the doctors can register and use the DBOQS, which has a basic menu for the doctor to use. Once the doctor has selected an item from the menu (Message input), the LINE Application will send a message request to the LINE platform that provides the DBOQS service for checking the message event received from the LINE Application. Once completed, they will send a message reply to the LINE platform, and the LINE platform will send the message back to the LINE Application for the doctor to read the message. For the administrator who developed the model, the model obtained from Section 3.2 will be the model that will be used in the DBOQS. Moreover, information with regard to the treatment program will be inputted as the information given to the user.

The DBOQS or the LINE Messaging API was collaboratively applied with CNN through the Flask API for assisting with the diagnosis of DR-TB. Once DBOQS receives an X-ray image that is sent by the user, the DBOQS will transmit the image to the DBOQS server’s webhook, and then the image is sent to the deep learning model. The deep learning models will predict the disease in the submitted image. Drug recommendations for predictable diseases are then extracted, and predictable disease outcomes and explanations are sent in text via the LINE Messaging API and LINE platform. This system can work for 24 h per day in real-time.

## 4. Computational Results

In this section, the computational results will be divided into two main parts, which are: (1) The test to reveal the effectiveness of the proposed methods, and (2) the result of developing DBOQS for the drug response classification.

### 4.1. Revealing the Effectiveness of the Proposed Models

This section will be divided into two sub-sections, which are: (1) Comparing the efficiency among all proposed models, in order to have multi-class classification and (2) comparing the classification of the proposed model with an existing method that has binary classification. All comparisons will use the same dataset, as explained in Section 3.1.

#### 4.1.1. Comparing the Effectiveness of the Proposed Models

All 60 methods will be evaluated in multi-class classification quality using the classification of four classes, and the results of all 60 methods are shown in Table 6.

From the results obtained in Table 6, we can see that M-12 obtained the best results of all key performance measurements. The confusion matrices of the classification are shown in Figure 14. The confusion matrices reveal that when employing a multi-class classification model to categorize DS-TB, the majority of incorrect results fall into the MDR-TB class, whereas the other two classes have the same percentage of incorrect results. There is no major difference between the remaining classification ambiguity. Using the binary classification approach, DS-TB has the highest proportion of incorrect predictions when compared pairwise with other classes. However, the classification of MDR-TB and XDR-TB is the most perplexing, with 15.43% of incorrect classifications.

From Table 7, using no preprocessing technique yields a worse solution than using all the preprocessing techniques (10.05%). Using all the CNN architectures in the ensemble yields a result that is 6.42% better than using the same CNN architecture in the ensemble. Finally, using AMIS-EDL yields a result that is 1.17% better than using UMA and MV as the decision fusion strategy. Utilizing all data augmentation approaches and data normalization in conjunction with all CNN architectures (heterogeneous) and AMIS-EDL as the decision fusion mechanism will be our plan for the next experiment.

#### 4.1.2. Comparing the Effectiveness of the Proposed Model with Existing Methods

In this experiment, we perform a comparison of the best method found in Section 4.1 with the existing heuristics. The comparison will be performed as the binary classification, and the result is shown in Table 8, Table 9, Table 10 and Table 11.

Comparing the classification of DS-TB and DR-TB based on the results in Table 8, the AUCs of the proposed model are 43.13% and 21.39% better than those of Ureta and Shrestha [75], and of Karki et al. [11]. Using F-measure as the criterion, the proposed model is 1.09% better than the model of Tulo et al. [76]. When classifying between DS-TB and DR-TB, we can conclude that the suggested method is 30.01% more accurate than the alternative methods. Comparing the classification results of DR-TB and MDR-TB between the proposed approaches of Jaeger et al. [24] and Tulo et al. [12], utilizing all KPIs, the proposed method provides a superior answer by 2.52–46.27% (using the information provided in Table 9). The classification results of DR-TB and XDR-TB utilizing the data presented in Table 10 demonstrated that the proposed approach is 1.71–5.52% superior to the method of Tulo et al. [12]. Finally, Table 11 reveals that the proposed approaches produce solutions that are 2.43% to 6.67% more accurate than the method of Tulo et al.

### 4.2. The DBOQS Result

DBOQS has been developed using the concept explained in Section 3.3, and the model developed in Section 4.1, which is model AD-10, will be used as the model for DBOQS. The treatment program suggested is the WHO recommendation program, as we mentioned in Section 1. An example of the Q and A results of DBOQS is shown in Figure 15.

We perform the experiment in DBOQS, and input the CXR 443 images (randomly selected from the test and train dataset) to identify its accuracy and other performance measures; the results are shown in Table 12.

The results in Table 12 indicate that the proper categorization rate is 90.74%, which is not significantly different from the results in Section 4.1. DS, DR, MDR, and XDR had respective accurate answers of 90.38%, 90.08%, 91.30%, and 91.26%. It is demonstrated that each class has a classification accuracy of at least 90%, a level that is sufficient for clinicians to place their trust in the classifications.

In the next experiment, we let 30 doctors try our DBOQS and evaluate them via a questionnaire; the results of the questionnaire are presented in Table 13. The scores from the doctors were averaged. A score equal to 1 indicated that the doctors did not agree with the question; when the score was 10, this indicated that the doctor strongly agreed with the question.

Table 13 shows that users rated the DBOQS 9.51 out of 10 for its response time, suggesting that they found it to be extremely helpful. With a mean score of 9.58, people agreed that the DBOQS provides the correct answer when asked about drug resistance. With a score of 9.97 and 9.86, respectively, they believed that the DRCS would make it simpler to diagnose the drug reaction and lessen their burden. After the testing phase of the study, participants rated their trust in and willingness to utilize the DRCS as high as 9.51 and 9.97, respectively.

## 5. Discussion

The developed approach in this study is one of the first to classify TB patients into four distinct groups based on their responses to the treatment. DS-TB, DR-TB, MDR, and XDR are the subtypes that are currently known. Since there are many possible ways to classify a patient’s reaction to a medicine, a multi-class classification model was used. Two additional studies, Karki et al. [11] and Tulo et al. [12], also divide TB patients into four groups, but they achieve this using a binary classification system (BC). Every time BC is used, it will always provide a 0 or a 1 for the outcome, such as when comparing DS and DR. Furthermore, our algorithms can be classified into four classes simultaneously, making this classification model more user-friendly. Once we know the type of TB, we can more quickly plan the patient’s treatment schedule.

Additionally, the AMIS-EDL model was created to enhance the quality of the solutions provided using pre-existing models. Computational results demonstrate that AMIS-EDL outperforms the current approaches, including the ones proposed by Karki et al. [11] and Tulo et al. [12]. This is due to the fact that the AMIS algorithm, as developed by Pitakaso et al. [61], has been shown to be effective. For example, it can obtain better results than the genetic algorithm and the differential evolution algorithm, two of the most popular current heuristics. The suggested model outperforms the conventional DL method, which relies on the unweighted model average (UMA) and majority voting (MV) by a profit of 31.25%. This follows the reasoning of Gonwirat and Surinta [91], who similarly concluded that metaheuristics, such as AMIS, can enhance the quality of solutions obtained using DLs employing the conventional UMA and MV.

The user has confidence in the DRCS’s ability to generate a credible result of 95.1%, according to the computational result. As a result of the DBOQS’s role, the medical staff usually do not give it a sufficient amount of chance for employment in medical care due to low trust and the appearance of unreliability. Medical staff were overstretched due to the increasing need for hospital care during the COVID-19 pandemic. The accuracy and dependability of a DBOQS are key factors in increasing the likelihood that medical professionals will choose to utilize it as an aid. This is why our system has a good chance of being adopted and utilized by medical experts to aid in the diagnosis and treatment planning of tuberculosis.

As we know from the results, the DRCS has more than 90% accuracy, and it can improve precision from traditional and existing methods by 1.17% to 43.43%. Therefore, when the accuracy is high and the workload is high, it would be good for the medical staff to use the DRCS for reducing their workload, and it would increase the service quality of the medical staff, as well.

## 6. Conclusions and Outlook

This article develops effective methods for classifying DS-TB, DR-TB, MDR, and XDR patients. Using ensemble deep learning, the desired outcome has been predicted. We selected four small and effective CNN architectures for deep learning; namely, MobileNetV2, EfficientNetB7, NASNetMobile, and DenseNet121. In addition, image augmentation and a normalization technique were incorporated into the model to improve the quality of the proposed methods. Finally, unweighted ensemble averaging, majority voting, and AMIS-EDL were used to discover the optimal weight parameters for the ensemble deep learning system. The proposed methods were compared to the performances of several types of methods addressed in previous studies.

The computational findings demonstrate that, among the deep learning architectures offered, EfficientNetB7 offers the most effective solution. It improves the solution by 1.76–2.41% compared to the others. The best architecture for classifying the DR is the one that combines all the architectures into a single model. The optimum architecture for classifying DR utilizes all the CNN architectures inside a single classification model. Using multiple CNN architectures can enhance the quality (accuracy) by 5.21–7.75%. The preprocessing techniques increase the quality of the answer by an average of 10.41% when compared to methods that do not use preprocessing. In comparison to other fusion methods, AMIS can improve the solution quality by 1.17%.

In comparison to the method described by Karki et al. [11], the solution quality can be enhanced by 31.25%. Compared to the other approaches proposed in a previous study, the proposed method can provide solutions that are 2.87–7.15% better than Tulo et al. [12], 48.58% better than Jaeger et al. [24], 53.16% better than Kovalev et al. [77], and 43.13% better than Ureta and Shrestha [75]. We can conclude, based on this result, that the proposed method surpassed all of the other methods proposed in the literature in terms of producing more exact classifications. The computational results demonstrate that the DBOQS developed for use in the DRCS can predict drug responses with an accuracy of more than 90% in every group of drug response, and the overall accuracy is 90.74%. Overall, with a result of 9.51 points out of 10 points, the clinicians who used the developed system had faith in it, and 99.7% of the doctors planned to keep using it as a secondary diagnostic tool in the future.

It is necessary to further develop the DRCS in order that it can: (1) Classify the various forms of drug resistance; (2) recommend a treatment regimen; (3) monitor the patient’s progress after beginning the recommended regimen; and (4) collect data from the user in order to analyze and recommend a subsequent treatment plan. Despite this, data mining and two-way communication DBOQSs are vital for the DRCS to make informed decisions and to provide meaningful advice to medical professionals when necessary.

## Figures and Tables

**Figure 1 diagnostics-12-02980-f001:**
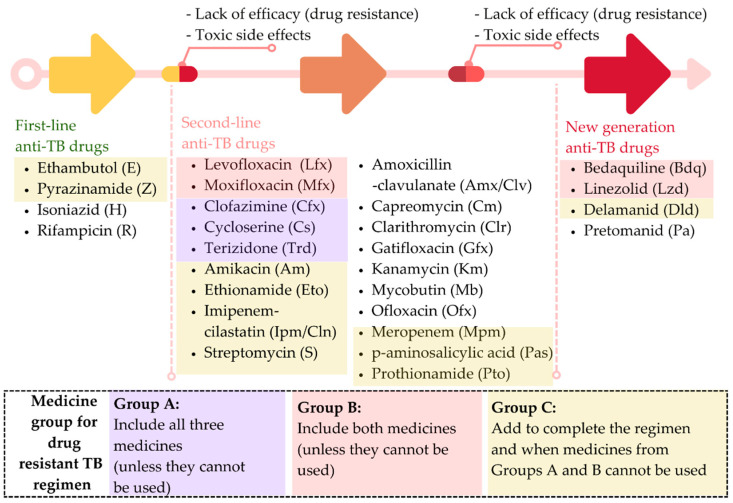
Treatment strategy for tuberculosis (adapted from World Health Organization [3]).

**Figure 2 diagnostics-12-02980-f002:**
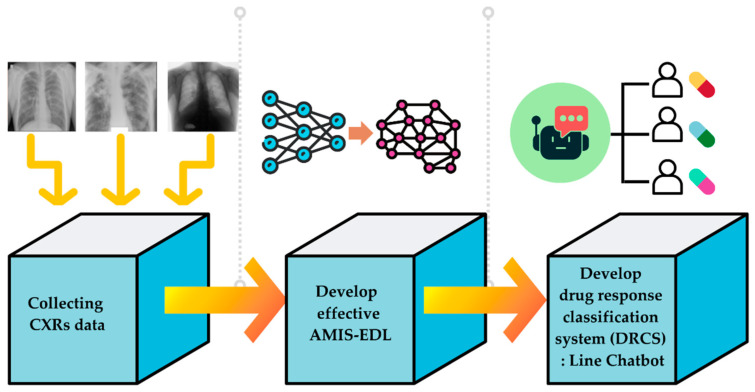
The development of drug response classification system (DRCS) (DBOQS).

**Figure 3 diagnostics-12-02980-f003:**
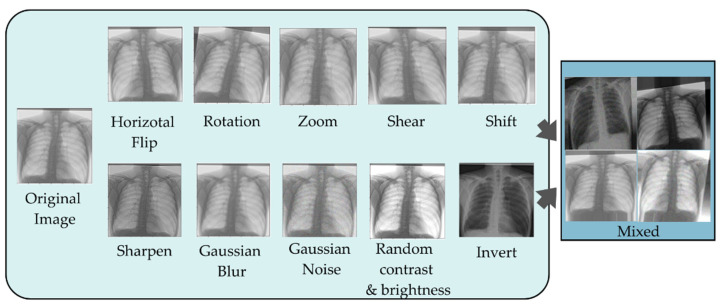
Image augmentation example.

**Figure 4 diagnostics-12-02980-f004:**
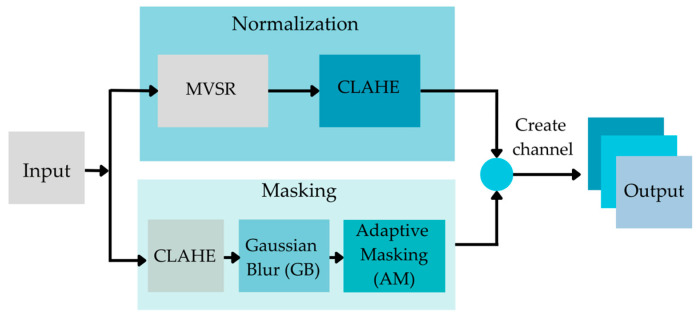
Framework of the data normalizing model.

**Figure 5 diagnostics-12-02980-f005:**
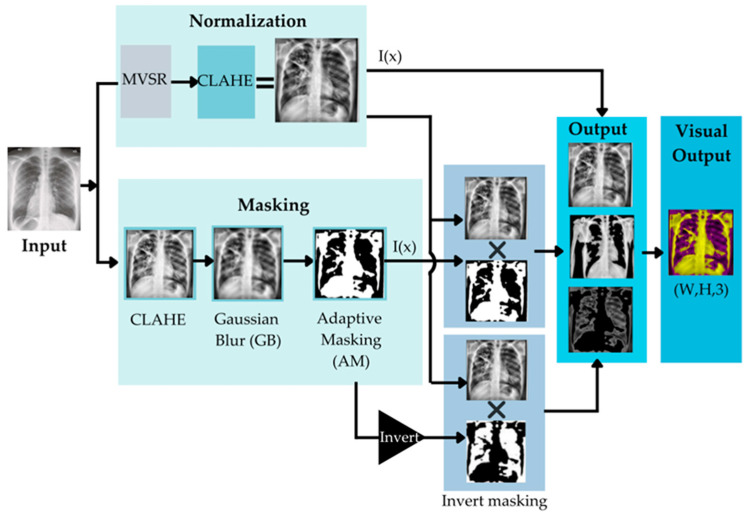
A flow diagram to illustrate image pre-processing steps to generate the input of a CNN model.

**Figure 6 diagnostics-12-02980-f006:**
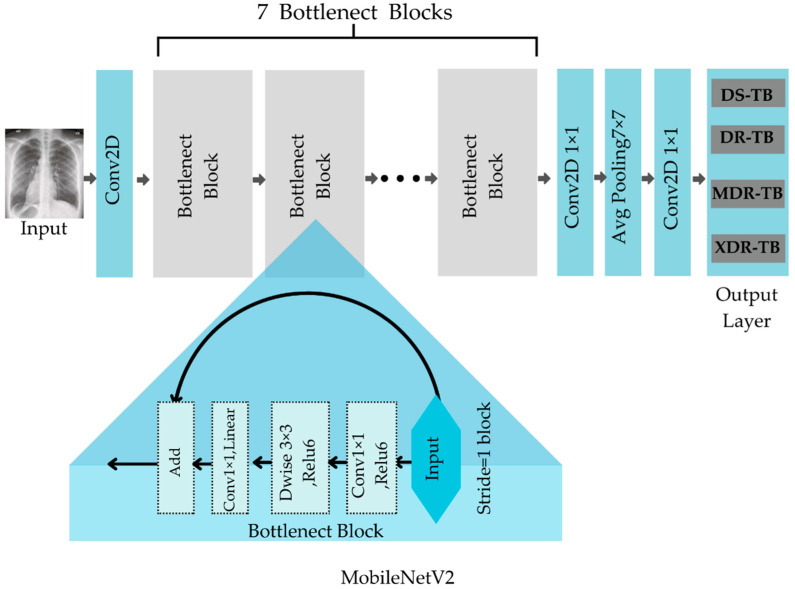
Basic structure of MobileNetV2.

**Figure 7 diagnostics-12-02980-f007:**
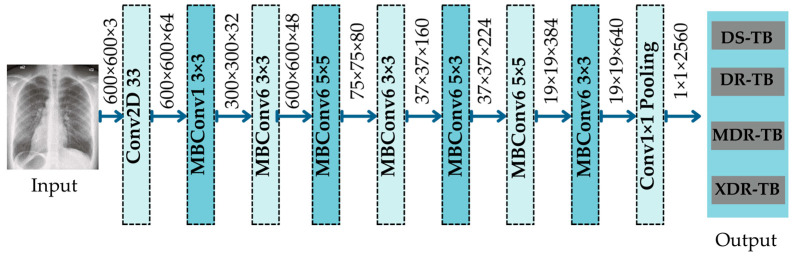
Architecture of EfficientNetB1.

**Figure 8 diagnostics-12-02980-f008:**
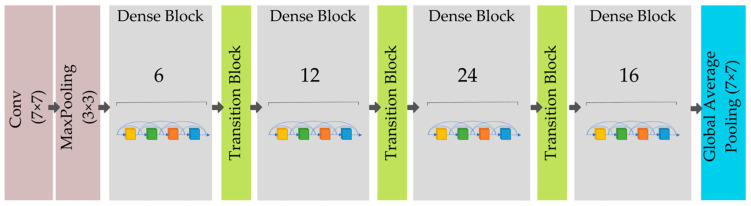
Example of a deep DenseNet with three dense blocks.

**Figure 9 diagnostics-12-02980-f009:**
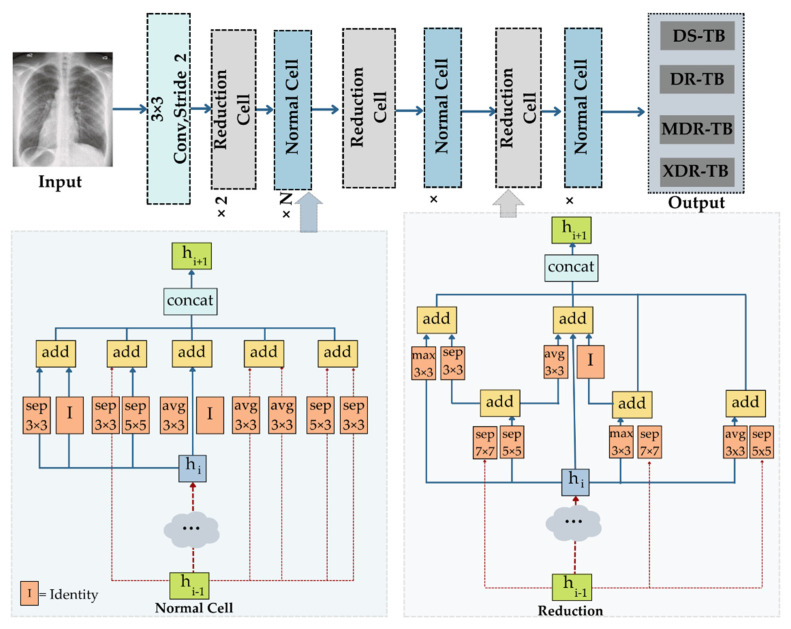
The NASNetMobile architecture.

**Figure 10 diagnostics-12-02980-f010:**
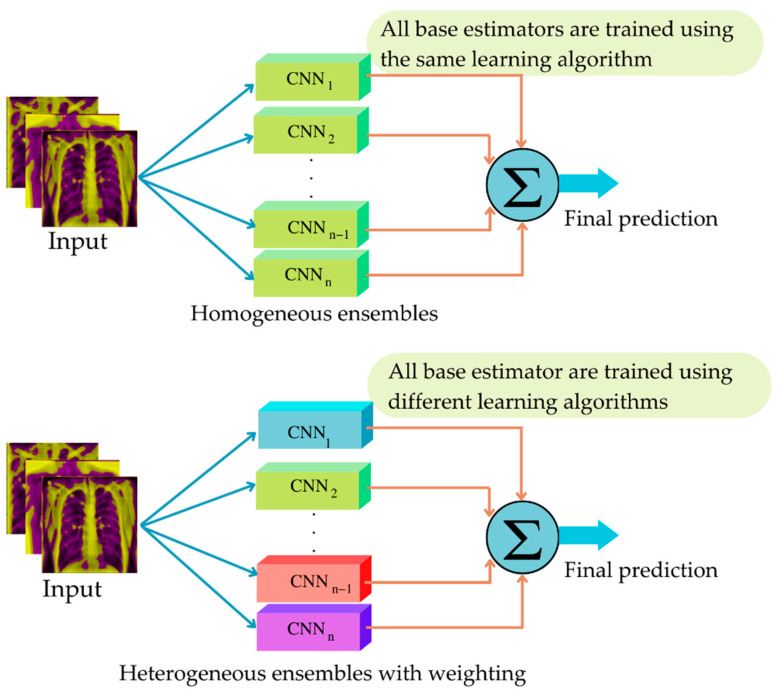
Structure of the EDL using heterogeneous and homogenous manners.

**Figure 11 diagnostics-12-02980-f011:**
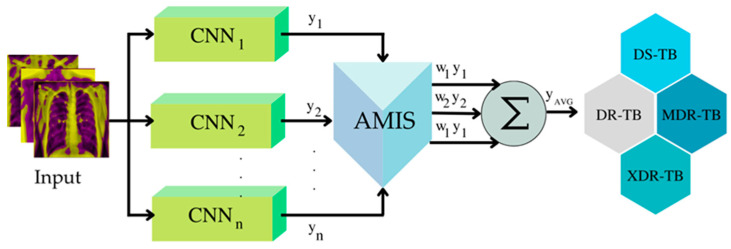
Framework of how to use weight generation to find the optimal weight of ensemble DL.

**Figure 12 diagnostics-12-02980-f012:**
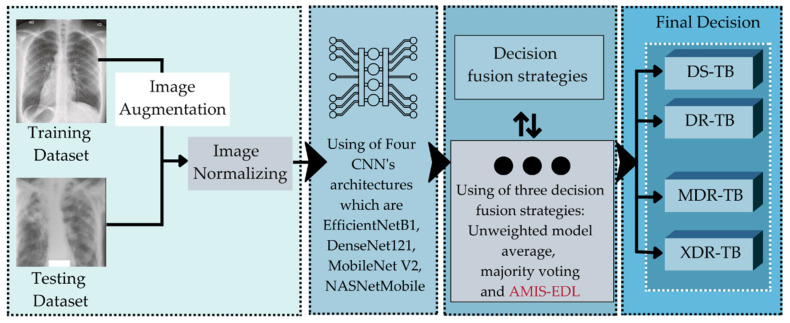
The framework of the proposed model used to classify the drug response of the TB patients.

**Figure 13 diagnostics-12-02980-f013:**
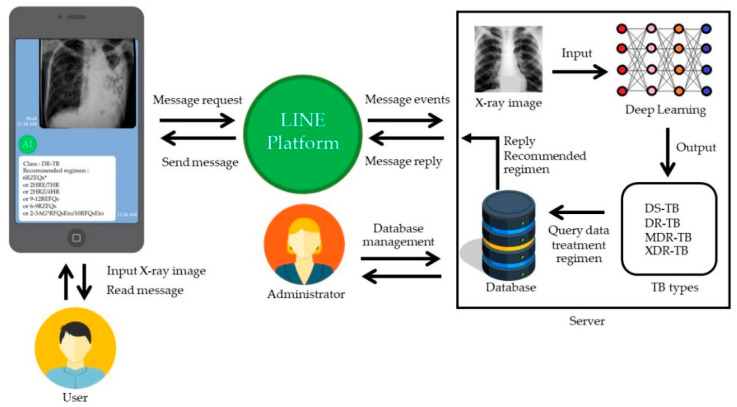
The design framework of DBOQS for the case study.

**Figure 14 diagnostics-12-02980-f014:**
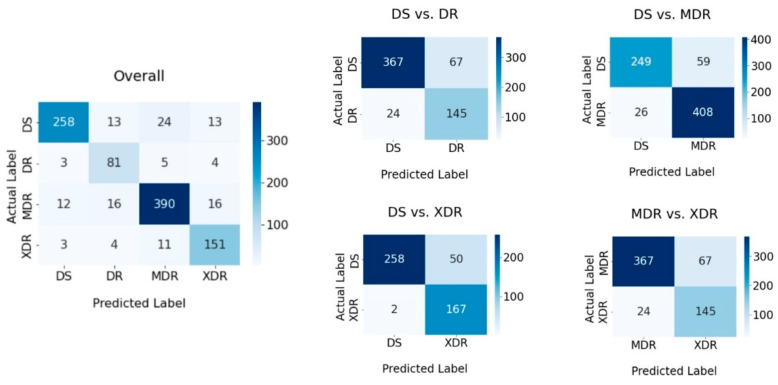
Confusion matrices of the classification result shown in Table 4.

**Figure 15 diagnostics-12-02980-f015:**
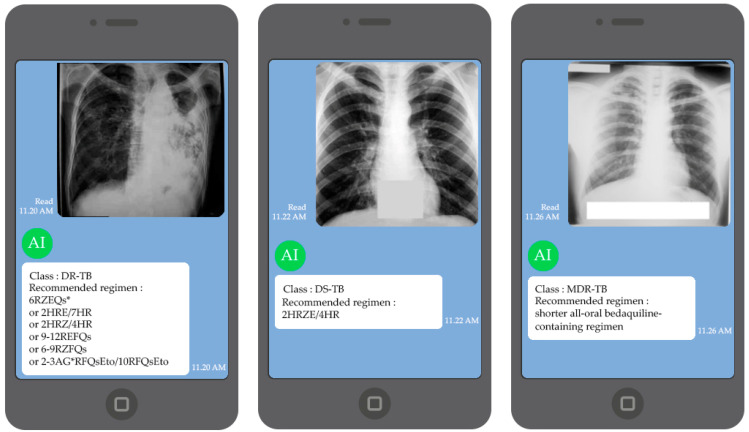
Q and A results of DBOQS.

**Table 1 diagnostics-12-02980-t001:** Number of data of each category.

	DS-TB	DR-TB	MDR-TB	XDR-TB
Train set	1299	375	1653	688
Test set	308	93	434	169
Total	1607	468	2087	857

**Table 2 diagnostics-12-02980-t002:** Details of the compared method in the existing research.

Research	Classes	Features	Region in CXR	AUC	Accuracy F-Measure	Accuracy
Ureta and Shrestha [76]	DS vs. DR	CNN	Whole	67.0	-	-
Tulo et al. [77]	DS vs. DR	Shape	Mediastinum + Lungs		93.6%	
Jaeger et al. [24]	DS vs. MDR	Texture, Shape, and Edge	Lung	66%	61%	62%
Kovalev et al. [78]	DS vs. DR	Texture and Shape	Lung	-	-	61.7
Tulo et al. [12]	DS vs. MDR	Shape	Mediastinum + Lungs	87.3	82.4	82.5
Tulo et al. [12]	MDR vs. XDR	Shape	Mediastinum + Lungs	86.6	81.0	81.0
Tulo et al. [12]	DS vs. XDR	Shape	Mediastinum + Lungs	93.5	87.0	87.0
Karki et al. [11]	DS vs. DR	CNN	Lung excluded	79.0	-	72.0
Proposed method	DR vs. DS	Ensemble CNN	Whole	-	-	-
Proposed method	DS vs. MDR	Ensemble CNN	Whole	-	-	-
Proposed method	DS vs. XDR	Ensemble CNN	Whole	-	-	-
Proposed method	DR vs. MDR vs. XDR	Ensemble CNN	Whole	-	-	-

**Table 3 diagnostics-12-02980-t003:** Initial WP when D = 10 and *i* = 5.

NP\D	1	2	3	4	5	6	7	8	9	10
1	0.64	0.82	0.61	0.93	0.40	0.99	0.70	0.68	0.36	0.08
2	0.58	0.28	0.99	0.79	0.26	0.60	0.09	0.20	0.09	0.85
3	0.48	0.90	0.04	0.95	0.85	0.29	0.09	0.08	0.93	0.49
4	0.34	0.69	0.91	0.12	0.33	0.97	0.20	0.73	0.14	0.61
5	0.88	0.25	0.28	0.32	0.51	0.75	0.06	0.73	0.10	0.93

**Table 4 diagnostics-12-02980-t004:** Updated role of heuristics information.

Variables	Update Method
$N_{bt}$	Sum of work packages (WPs) that have selected IB b from iteration 1 through iteration *t*
$A_{bt}$	The average objective value selected by IBs $\left(\frac{\sum_{i=1}^{N_{bt}}f_{}}{N_{bt}}\right)$
$I_{bt}$	$I_{bt}=I_{bt-1}+G$ When $G=\{\begin{array}{c} 1if\wedge onlyifblackboxbhastheglobalbestsolutionatiterationt\\ 0otherwise \end{array}$
$B_{j}^{gbest}$	Update current global best WP
$B_{hj}^{pbest}$	Update IB’s best WP
$R_{ijt}$	Pick a random positional value across all WPs

**Table 5 diagnostics-12-02980-t005:** Conclusions of 36 proposed models for drug response classification of TB patients.

Model	Preprocessing		CNN Architectures				Decision Fusion Strategy		
	Data Augmentation	Data Normalizing	EfficientNetB7	DenseNet121	NASNetMobile	MobileNetV2	UMA	MV	AMIS-EDL
N-1			✓						
N-2			✓					✓	
N-3			✓						✓
N-4				✓			✓		
N-5				✓				✓	
N-6				✓					✓
N-7					✓		✓		
N-8					✓			✓	
N-9					✓				✓
N-10						✓	✓		
N-11						✓		✓	
N-12						✓			✓
A-1	✓		✓				✓		
A-2	✓		✓					✓	
A-3	✓		✓						✓
A-4	✓			✓			✓		
A-5	✓			✓				✓	
A-6	✓			✓					✓
A-7	✓				✓		✓		
A-8	✓				✓			✓	
A-9	✓				✓				✓
A-10	✓					✓	✓		
A-11	✓					✓		✓	
A-12	✓					✓			✓
D-1		✓	✓				✓		
D-2		✓	✓					✓	
D-3		✓	✓						✓
D-4		✓		✓			✓		
D-5		✓		✓				✓	
D-6		✓		✓					✓
D-7		✓			✓		✓		
D-8		✓			✓			✓	
D-9		✓			✓				✓
D-10		✓				✓	✓		
D-11		✓				✓		✓	
D-12		✓				✓			✓
AD-1	✓	✓	✓				✓		
AD-2	✓	✓	✓					✓	
AD-3	✓	✓	✓						✓
AD-4	✓	✓		✓			✓		
AD-5	✓	✓		✓				✓	
AD-6	✓	✓		✓					✓
AD-7	✓	✓			✓		✓		
AD-8	✓	✓			✓			✓	
AD-9	✓	✓			✓				✓
AD-10	✓	✓				✓	✓		
AD-11	✓	✓				✓		✓	
AD-12	✓	✓				✓			✓
M-1			✓	✓	✓	✓	✓		
M-2			✓	✓	✓	✓		✓	
M-3			✓	✓	✓	✓			✓
M-4	✓		✓	✓	✓	✓	✓		
M-5	✓		✓	✓	✓	✓		✓	
M-6	✓		✓	✓	✓	✓			✓
M-7		✓	✓	✓	✓	✓	✓		
M-8		✓	✓	✓	✓	✓		✓	
M-9		✓	✓	✓	✓	✓			✓
M-10	✓	✓	✓	✓	✓	✓	✓		
M-11	✓	✓	✓	✓	✓	✓		✓	
M-12	✓	✓	✓	✓	✓	✓			✓

**Table 6 diagnostics-12-02980-t006:** Multi-Class Classification Result.

Methods	AUC	F-Measure	Accuracy	Methods	AUC	F-Measure	Accuracy
N-1	76.1	71.5	73.2	D-7	80.7	75.5	77.0
N-2	75.9	72.1	74.7	D-8	78.5	73.5	75.6
N-3	76.9	73.1	75.4	D-9	83.4	78.2	80.4
N-4	76.4	70.5	71.4	D-10	79.5	75.2	76.4
N-5	74.9	71.6	72.2	D-11	79.5	76.5	78.4
N-6	77.4	73.8	75.6	D-12	81.1	75.5	77.5
N-7	72.5	69.4	70.8	AD-1	85.4	80.7	83.2
N-8	74.8	70.4	71.2	AD-2	85.6	82.3	84.6
N-9	76.5	71.4	74.2	AD-3	86.9	83.5	85.2
N-10	77.9	73.5	75.2	AD-4	84.6	82.5	83.4
N-11	77.3	72.5	75.3	AD-5	83.8	80.3	81.3
N-12	76.3	73.1	74.2	AD-6	83.0	81.0	82.1
A-1	78.7	75.7	77.6	AD-7	83.4	81.8	82.1
A-2	80.2	77.6	78.3	AD-8	83.2	81.0	82.4
A-3	80.6	78.1	79.0	AD-9	84.5	80.7	81.4
A-4	77.5	71.3	74.4	AD-10	84.1	81.6	82.4
A-5	76.3	71.9	73.2	AD-11	84.5	82.2	82.3
A-6	76.8	73.5	75.2	AD-12	86.7	87.5	82.3
A-7	76.4	72.5	74.0	M-1	84.5	82.3	83.5
A-8	77.8	73.5	74.5	M-2	84.5	81.2	82.8
A-9	78.3	75.0	76.7	M-3	84.8	83.2	84.6
A-10	78.4	74.9	75.3	M-4	85.3	82.3	83.6
A-11	79.3	75.5	76.1	M-5	86.5	83.5	84.5
A-12	78.9	75.0	76.6	M-6	84.9	82.7	83.6
D-1	82.4	77.7	78.8	M-7	86.5	84.1	85.4
D-2	80.2	77.9	79.0	M-8	86.4	81.6	82.4
D-3	84.4	81.2	82.0	M-9	87.1	85.2	86.6
D-4	80.1	77.8	79.7	M-10	88.4	85.8	86.1
D-5	81.4	78.7	79.4	M-11	88.5	85.5	86.2
D-6	82.5	79.4	80.6	M-12	88.9	85.6	87.6

All experiments are conducted, and the results presented in Table 6 are summarized in Table 7.

**Table 7 diagnostics-12-02980-t007:** Percentage average accuracy value of using different strategies of the data presented in Table 6.

Preprocessing Techniques				CNN Architectures					Decision Fusion Strategies		
No Preprocessing	Data Augmentation	Data Normalizing	Use All	EfficientNetB7	DenseNet121	NASNetMobile	MobileNetV2	Use All	UMA	MV	AMIS-EDL
74.96	80.5	81.7	**82.7**	80.6	79.2	78.7	79.4	**84.8**	78.7	78.7	**80.1**

**Table 8 diagnostics-12-02980-t008:** The accuracy of the proposed methods with the existing heuristics to classify DR-TB and DS-TB.

Methods	Classes	Features	Region in CXR	Accuracy		
				AUC	F-Measure	Accuracy
Ureta and Shrestha [76]	DS vs. DR	CNN	Whole	67.0	-	-
Tulo et al. [77]	DS vs. DR	Shape	Mediastinum + Lungs	-	93.6	-
Kovalev et al. [78]	DS vs. DR	Texture and Shape	Lung	-	-	61.7
Karki et al. [11]	DS vs. DR	CNN	Lung excluded	79.0	-	72.0
Proposed method	DS vs. DR	Ensemble CNN	Whole	97.9	93.8	94.8

**Table 9 diagnostics-12-02980-t009:** The accuracy of the proposed methods with the existing heuristics to classify DS-TB and MDR.

Methods	Classes	Features	Region in CXR		Accuracy	
				AUC	F-Measure	Accuracy
Jaeger et al. [24]	DS vs. MDR	Texture, Shape, and Edge	Lung	66	61	62
Tulo et al. [12]	DS vs. MDR	Shape	Mediastinum + Lungs	87.3	82.4	82.5
Proposed method	DS vs. MDR	Ensemble CNN	Whole	89.5	88.0	88.5

**Table 10 diagnostics-12-02980-t010:** The accuracy of the proposed methods with the existing heuristics to classify DS-TB and XDR.

	Classes	Features	Region in CXR		Accuracy	
				AUC	F-Measure	Accuracy
Tulo et al. [12]	DS vs. XDR	Shape	Mediastinum + Lungs	93.5	87.0	87.0
Proposed method	DS vs. XDR	Ensemble CNN	Whole	95.1	88.7	89.1

**Table 11 diagnostics-12-02980-t011:** The accuracy of the proposed methods with the existing heuristics to classify MDR and XDR.

	Classes	Features	Region in CXR		Accuracy	
				AUC	F-Measure	Accuracy
Tulo et al. [12]	MDR vs. XDR	Shape	Mediastinum + Lungs	86.6	81.0	81.0
Proposed method	MDR vs. XDR	Ensemble CNN	Whole	88.7	82.5	84.9

**Table 12 diagnostics-12-02980-t012:** The DBOQS accuracy results.

	Number of Input Images	Number of Correct Classification	%Correct Classification	Number of Wrong Classification	%Wrong Classification
DS	104	94	90.38	10	9.62
DR	121	109	90.08	12	9.92
MDR	115	105	91.30	10	8.70
XDR	103	94	91.26	9	8.74
Total	443	402	90.74	41	9.26

**Table 13 diagnostics-12-02980-t013:** Questionnaire result from 30 doctors who were trying to use DRCS.

Questionnaire on Images	Level of Agreement (Strong Agreement Level Is 10)
DBOQS has fast responses.	9.51
DBOQS can classify the drug response correctly.	9.58
DBOQS may simplify the drug response diagnosis process.	9.97
DBOQS can reduce your workload.	9.86
You will use the DBOQS to aid in your drug response diagnosis.	9.74
What score will you assign the DBOQS with regard to your trust level?	9.51
What score will you assign the DBOQS with regard to your preferences?	9.97

## Data Availability

Not applicable.

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
