# Peer review of "Embedded AMIS-Deep Learning with Dialog-Based Object Query System for Multi-Class Tuberculosis Drug Response Classification"

_diagnostics, 2022, doi:10.3390/diagnostics12122980_

Round 1
Reviewer 1 Report
The authors used two types of image preprocessing methods, four CNN architectures, and three decision fusion methods to classify DS-TB, DR-TB, MDRTB, and XDR-TB. They also develop DBOQS to to assist medical professionals in diagnosing DR-TB. The authors found a way to improve the accuracy of classification. The results are good and interesting. The structure of the manuscript is fairly clear, but some typos are in the text. It can be published with minor revision.
1. Use the first author's name instead of the reference number. For example,
page 3: Machine learning was ultimately used by [12] to categorize..
page 5: Before feeding the images into the CNN model, [34] and [35] applied cropping...
2. The resolution of figures is not good enough for publication.
3. In Table 12, the Number of Wrong Classification and the %Wrong Classification is not consistent. Why?
4. In Fig. 12, why is the image augmentation only applied to the testing datasets, not the training datasets? In page 10, the authors stated "By creating synthetic datasets, data augmentation aims to increase the quantity and variety of training data."
Author Response
Thank you for your valuable recommendations. We have tried our best to resolve all issues that concerning about your suggestion.
Comment |
Answers: |
1. Use the first author's name instead of the reference number. For example, · page 3: Machine learning was ultimately used by [12] to categorize.. · page 5: Before feeding the images into the CNN model, [34] and [35] applied cropping |
Recheck through the whole paper and revised all mistaken. |
2. The resolution of figures is not good enough for publication. |
The resolution of all figures was adjusted to 330 dpi (high definition). |
3. In Table 12, the Number of Wrong Classification and the %Wrong Classification is not consistent. Why? |
I greatly appreciate your insightful feedback. The error was made by us, and it has been rectified. |
4. In Fig. 12, why is the image augmentation only applied to the testing datasets, not the training datasets? In page 10, the authors stated "By creating synthetic datasets, data augmentation aims to increase the quantity and variety of training data." |
We appreciate the valuable feedback; nonetheless, the issue is once again ours; we have corrected Figure 12 and ensured that all Figures and text are uniform across the whole manuscript. Thank you for allowing us to recognize our flaws and mistakes. |

Reviewer 2 Report
The authors provide a novel deep learning-based system for multi-class tuberculosis drug response classification. While the results look promising, I have some doubts and hope authors could provide some clarifications.
1. There are some typos and grammar errors in the writing. Please revise them. The format is not organized well, should be revised as well.
2. Figures are not clear for people to read. Please update.
3. Equation (11) has a typo. Some of the equations’ math symbols are not correct, please revise.
Author Response
Thank you very much for your valuable comments. We have tried our best to fulfill the recommendations from the reviewers.
Comment |
Answers: |
1) There are some typos and grammar errors in the writing. Please revise them. The format is not organized well, should be revised as well. |
The paper was sent to MDPI language and layout service since 17 November 2022 and the certificate of English and layout editing was attached below. |
2) The resolution of figures is not good enough for publication. |
The resolution of all figures was adjusted to 330 dpi (high definition). |
3) Figures are not clear for people to read. Please update. |
Font size in all figures were adjusted to readable size. |

Round 2
Reviewer 2 Report
The revision has improved the quality of the manuscript